# Toward Tightly Tuned Gene Expression Following Lentiviral Vector Transduction

**DOI:** 10.3390/v12121427

**Published:** 2020-12-11

**Authors:** Audrey Page, Floriane Fusil, François-Loïc Cosset

**Affiliations:** CIRI-Centre International de Recherche en Infectiologie, University of Lyon, Université Claude Bernard Lyon 1, Inserm, U1111, CNRS, UMR5308, ENS Lyon, 46 allée d’Italie, F-69007 Lyon, France; audrey.page1@ens-lyon.fr (A.P.); floriane.fusil@ens-lyon.fr (F.F.)

**Keywords:** lentiviral vectors, induction, transgene, signal, sensor, integration, promoter, synthetic biology

## Abstract

Lentiviral vectors are versatile tools for gene delivery purposes. While in the earlier versions of retroviral vectors, transgene expression was controlled by the long terminal repeats (LTRs), the latter generations of vectors, including those derived from lentiviruses, incorporate internal constitutive or regulated promoters in order to regulate transgene expression. This allows to temporally and/or quantitatively control transgene expression, which is required for many applications such as for clinical applications, when transgene expression is required in specific tissues and at a specific timing. Here we review the main systems that have been developed for transgene regulated expression following lentiviral gene transfer. First, the induction of gene expression can be triggered either by external or by internal cues. Indeed, these regulated vector systems may harbor promoters inducible by exogenous stimuli, such as small molecules (e.g., antibiotics) or temperature variations, offering the possibility to tune rapidly transgene expression in case of adverse events. Second, expression can be indirectly adjusted by playing on inserted sequence copies, for instance by gene excision. Finally, synthetic networks can be developed to sense specific endogenous signals and trigger defined responses after information processing. Regulatable lentiviral vectors (LV)-mediated transgene expression systems have been widely used in basic research to uncover gene functions or to temporally reprogram cells. Clinical applications are also under development to induce therapeutic molecule secretion or to implement safety switches. Such regulatable approaches are currently focusing much attention and will benefit from the development of other technologies in order to launch autonomously controlled systems.

## 1. Introduction

Since the early 1980s, viral vectors, and more particularly retroviral vectors (RVs), have been engineered as tools for gene delivery. Lentiviral vectors (LVs) are derived from specific species of lentiviruses, which form a genus in the *Retroviridae* family, and are nowadays widely used as tools to stably transfer or “transduce” their genetic material into the genome of target cells in many fields, from basic research to clinical applications. A characteristic making LVs valuable tools for gene transfer in somatic and germ-line cells, compared to other viral vectors, is their capacity to efficiently transduce non-dividing cells. In addition, like for many alternative enveloped viruses, LVs can be enveloped with heterologous virion surface glycoproteins, by a process called pseudotyping, which widens the range of cells they can efficiently transduce. For instance, pseudotyping lentiviral vectors with the glycoproteins from RD114 cat endogenous virus, baboon endogenous virus, or measles virus drastically increased the transduction efficiency of cells that were notoriously difficult to transduce, such as primary T or B lymphocytes, making gene transfer much more efficient [1,2,3]. Conversely, by carefully choosing the pseudotyping virion surface proteins, vector tropism may also be restricted to certain cell types; for example, through pseudotyping with Rabies virus glycoprotein or by incorporating on the vector particle surface antibodies directed against cell type specific surface compounds [4,5,6,7].

The production of LVs of the first generation relied on three plasmids encoding respectively the *Envelope* (*Env*) gene, all lentiviral ORFs except *Env*, and the expression cassette to be transferred that is associated to two viral long terminal repeats (LTRs) and to sequences required for viral RNA export, genome packaging and reverse transcription [8,9]. In the following LV generations, substantial modifications were introduced to enhance both the performance and the safety of gene transfer. Indeed, for the second generation of LVs, most retroviral “accessory” genes were removed, while in the third generation, the tat gene segment was also eliminated and the rev gene was expressed on a fourth plasmid [8,10,11]. Nevertheless, some safety issues still remained intrinsically associated to LV-mediated gene transfer, among which the risk of insertional mutagenesis. To reduce this risk, self-inactivated vectors (SIN), in which the enhancer/promoter sequence of the U3 sequence from the 3′LTR was deleted, have been engineered [12]. This deletion minimizes activation of genes located at close proximity of the vector integration site in the genome [8]. Initially, transgene expression was controlled by the LTRs, but in the later vector generations and obviously in the SIN vectors, different internal promoters and regulatory sequences were incorporated to express the transgene of interest [8,13]. Several promoters can be combined to express multiple transgenes from a single vector although this is limited by the packaging capacity of LVs (around 10 kb) and although this can lead to promoter interferences [13,14,15]. To overcome this hurdle, IRES sequences or 2A peptide sequences have been inserted in LVs, which induces internal cap-independent translation and ribosomal skipping mechanisms, respectively, and allows that only one promoter is required to express several transgenes (multicistronic vectors).

Of note, most of the promoters that have been used in LVs, such as the commonly used *cytomegalovirus* (CMV) minimal promoter, *Elongation factor 1a* (*EF1a*) promoter or *spleen focus-forming virus* (SFFV) promoter, are constitutive. Consequently, they do not allow regulated transgene expression, both quantitatively and timely, which is required for many applications, especially in the clinics. Indeed, regulated expression of the transgene under certain conditions, such as via exposure to exogenous or endogenous signal molecules or such as during specific periods, would exhibit several advantages. First, it may reduce the induction of immune responses against the transgenic protein. Second, a temporary pattern of promoter activity may reduce the potential activation of proto-oncogenes located in the vicinity of the inserted vector, hence decreasing the risk of malignancies. Finally, the possibility to tune not only ON or OFF but also to modulate the dose of transgene expression can be crucial to adjust it in response to variation of some host parameters. Thus, even more sophisticated technologies should be developed in order to generate LVs that can constantly monitor physiological parameters and adjust the output response via transgene expression and e.g., restore homeostasis.

To achieve a fine regulation, inducible LV systems must fulfill several criteria. On the one hand, the inducible promoter must display low background activity in the non-induced state, since a low transgene leakage activity might have dramatic consequences. On the other hand, under inducing conditions, the promoter activity must be high enough so that the transgene can exert its functions. In between these two extremes, the promoter activity might be or not adjusted depending on the amounts of inducer in order to tune the intensity of transgene expression. Obviously, these regulation properties should be compatible with the host cells and the regulation should be achieved without interfering with endogenous regulatory networks and cell physiology [16]. Finally, the inducible system must be compatible with potential limitations in the design of viral vectors, such as their packaging capacities.

Here we review the main inducible systems that have been invented for achieving transgene regulation upon LV-mediated gene transfer. The inducible systems can harbor promoters that are inducible by exogenous stimuli, such as small molecules (e.g., antibiotics) or temperature variations, or by the sensing of endogenous molecules, such as pathological markers (e.g., pro-inflammatory cytokines). Alternatively, transgene expression level might be indirectly adjusted by playing with modifications of inserted transgene loci, for example, by transgene excision. Overall, LVs that are inducible for transgene expression offer huge possibilities for both basic and translational research and are likely to become more and more widely used in the coming years.

## 2. Induction of Transgene Expression by External Stimuli

Inducible LVs might be controlled by exogenous stimuli that are provided at a precise timing and at a defined intensity. The inductor stimulus might regulate transgene expression either through the promoter or through the number of copies of the transgene. Several inductors have been investigated for transgene stimulation, such as temperature variations. Nevertheless, the most commonly used signals that have been implemented for inducible LVs are small molecules. Importantly, the small molecules chosen for transgene regulation should be bioavailable and biocompatible, meaning that their pharmacokinetic profiles should allow a tight regulation and that they should not be immunogenic.

### 2.1. Direct Control of Transgene Expression (Through Promoter Regulation)

Several small molecules have been considered to conditionally activate an inducible promoter, which can control transgene transcription. Among them, antibiotics are considered as good candidates because they are approved for clinical use, with absent or very low side effects but with desired pharmacokinetic profiles, and also because they are usually orthogonal to cell networks, i.e., they have no cross-talks with endogenous signaling cascades. Based on this approach, the most commonly used system is the Tet-system, which is activated by tetracycline or its analog doxycycline and which is compatible with lentiviral gene transfer technologies (Figure 1a).

This system has been described from the first time in 1992 [17]. It is based on two elements: (i) the *Escherichia coli* Tet repressor protein (TetR), which represses transcription by binding to the Tet operator (TetO) and (ii) tetracycline or its analogue, doxycycline. Upon drug addition, a conformational change is induced within the TetR, abolishing its binding to TetO, and hence transcription blockade. The activation domain of the *herpes simplex virus* VP16 protein is fused to the TetR (tetracycline controlled trans-activator, tTA) to induce gene expression when bound to a minimal promoter sequence derived from the human CMV promoter (Ptet) that is placed downstream TetO in absence of doxycycline (Tet-Off system). A four-amino acid modification within this activator was also introduced to create a reverse tTA (rtTA), which, conversely, requires tetracycline or doxycycline to bind Ptet, and thus to allow transgene transcription (Tet-On system). The choice of one of either system depends on the application and whether the system will be mainly studied in the induced vs. in the repressed state. For gene therapy and more generally for in vivo applications, the Tet-ON system would be more suited, since the Tet-Off system is dependent on the drug clearance rate and since, conversely, maintenance of the off-state would require chronic doxycycline administration to suppress gene expression.

The Tet-system presents numerous advantages for inducible gene expression. It is dose-dependent and fully reversible. Furthermore, it relies on the use of well-tolerated antibiotic drugs that can be easily administered. Indeed, these antibiotics can be either intravenously injected but they can also be added in the drinking water or in diet [18], without affecting host physiology. Nevertheless, their use is still associated with side effects, such as the emergence of antibiotic resistance [19]. Another issue related to this system is the immunogen and toxic potential of tTA and rtTA, which is likely a consequence of the VP16 domain. Several improvements of this system have been performed over the past decades to improve the performance of the system and to reduce side effects, notably by further engineering the rtTA. For instance, the rtTA2s-M2 transactivator was obtained by mutagenesis and does not contain the VP16 domain anymore. This construct exhibits improved stability, specificity and inductility [20,21]. In addition to trans-activator optimization, the tet-responsive promoters have also focused much attention to enhance properties of the system, leading to variants with reduced leakage and higher dynamic range of expression either by truncating the CMV minimal promoter in Ptet, by using the HIV-1 long terminal repeat promoter instead of CMV in Ptet, or by random mutations in Ptet [22,23,24].

In the meantime, the Tet-system has been successfully implemented for modulating gene expression upon transduction by LVs in order to regulate transgene expression in a large range of cells [20,25,26]. Initially, a “two-vectors” approach was developed, with one vector encoding the tetracycline response element fused to a promoter followed by the gene of interest and a second one encoding the transcriptional trans-activator or the trans-repressor. One hurdle associated to this strategy is the need to select cells that express the trans-activator or the trans-repressor, since it can lead to heterogeneity in cellular responses in absence of selection. Furthermore, the integration sites can also influence both the expression of the trans-activator or the trans-repressor and the transgene inducibility, and so the overall regulation performances. With two LVs, each carrying a part of the regulation system, the risk of insertional mutagenesis is also higher, which could be particularly problematic in terms of biosafety.

To circumvent these issues, this “two-vectors” approach was replaced by more advanced “all-in-one” LVs containing both the tetracycline regulatory element and the inducible expression cassette within the same construct. Initially, rTA or rtTA were controlled by weak constitutive promoters, such as the *phosphoglycerate kinase 1* promoter [27] or the *elongation factor 1a* short promoter [28], and the inducible transgene was placed after a Tet response element (TRE). Several construct configurations were tested, with the TRE promoter inserted either upstream [13,29] or downstream [30] the constitutive promoter. Several promoter architectures were compared in order to decrease promoter cross-talks and this showed that a head-to-head orientation was better than the tail-to-tail or the head-tail organizations [15]. Nevertheless, interferences between the constitutive promoter driving rTA or rtTA and the TRE promoter were still observed, leading to increased background levels. To reduce basal noise, auto-regulated LVs have been engineered. In such constructs, the rTA or rtTA are controlled by the TRE promoter and placed after IRES or T2A sequences [31,32,33]. Other auto-regulated configurations, in which two CMV promoter are placed under the regulation of the same TRE have been successfully designed [26]. In these auto-regulated constructs, the leakage of expression of the rTA or rtTA was sufficient for amplification of the system. In addition, the background expression levels were much decreased, and the kinetics of induction were improved. Tet auto-regulated LVs have also been shown to be operational in vivo, with no decrease of expression after several rounds of doxycycline induction, paving the way for future clinical applications [33]. Numerous cell lines and primary cells from rodents to humans have already been transduced by LVs containing the Tet-regulation system [13,34,35,36,37].

In parallel, orthogonal promoters that can respond to other signal molecules, such as cumate [38] or 6-hydroxy-nicotine [39], have been developed. Likewise, two novel antibiotic-regulatable systems based on the Tet-system principle have focused much attention. Indeed, streptogramin [40] and macrolide [41] responsive promoters have been engineered and successfully incorporated in LVs for transgene induction [42,43]. Interestingly, antibiotic-sensitive promoters have been combined to create Boolean logical gates [44,45]. Such complex regulation loops have not yet been incorporated in LVs although they represent promising tools for the construction of artificial gene networks, which would be invaluable for numerous applications in gene therapy and tissue engineering. Alternatively, food additives have also been considered as potent signals for induction of transgene expression. Indeed, like for antibiotics, they are biocompatible, inert and they can be easily orally ingested. Several food additives have been tested for transgene induction, such as caffeine [46], menthol [47], phloretin [48], spearmint [49], and vanilla derivatives [50,51]. However, undesired activation of such promoters might occur, as the presence of additives in food is not always noticed. However, the doses required for promoter activation might also not be physiologically reachable. The study of such regulation systems has been limited so far to transfection experiments and have not yet been translated in LV constructs.

Other signals may be used as inducers, such as light or temperature. They have mainly been studied in transfection assays but not in LV-mediated transduction set up until recently, with a recently developed LV construct containing the heat shock protein (HSP) 70 promoter and TRE promoter. HSP promoters carry heat response elements that can detect temperature variations and a switch in temperature from 37 °C to 43 °C could increase transgene expression [52]. However, most temperature inducible systems respond to large degrees of temperature variation, which makes them less suitable for in vivo applications. Interestingly, while the in vivo range of temperature variation is not that substantial, localized temperature variations may be induced by microwaves. However, such inducers present important drawbacks, especially in terms of orthogonality, since ambient light and external temperature can influence activation. Thus, progresses remain to be achieved before such systems can become widely used in vivo. Alternatively, ionizing radiations have been shown to activate the *early growth response* gene (*EGR1*) promoter via a consensus sequence. This opens the path for transgene activation by the radiation field to enhance treatment efficacy by local co-delivery of therapeutic molecules [53,54].

### 2.2. Indirect Control of Transgene Expression

In parallel, transgene expression can indirectly be adjusted by playing with the number of transgene copies or with the structure of the transgenic cassette into host genome, as discussed here.

Indeed, an obvious possibility to restrict the intensity of transgene expression is to tune the efficiency of gene transfer by using different MOI (multiplicity of infection) during transduction of target cells. While this will not allow a temporal control, this may spatially modulate the intensity of expression. For instance, transgenic protein gradients have been created in rat sciatic nerves by implementing a spatial MOI gradient during cell transduction [55].

Alternatively, the transgene copy number integrated in host genomic DNA after LV transduction can be modulated by inducible excision of the transgenic cassette (Figure 1b) [56,57]. This may be of specific interest in terms of safety. Indeed, through its insertion between two LoxP (locus of X-over P1) sites in a LV construct, a transgene expression cassette can be removed upon exposure to Cre recombinase, which can rapidly stop transgene expression. The Cre/LoxP system can also be used in the opposite approach to activate transgene expression. In this specific application, a silencing sequence, such as stop codon or a poly(A) signal, is inserted between the LoxP sites before the transgene in the LV construct. Following Cre recombinase treatment, this silencing sequence is removed, which allows transgene expression [56,58]. Note that for these applications, the Cre recombinase can be provided by viral transduction (i.e., using adenoviral or lentiviral vectors).

Finally, the expression of a transgenic protein may also be stopped by direct killing of transduced cells. This can be achieved by using several strategies that rely on the insertion of a suicide gene within the same construct than that of the transgene. Among them, the inducible *Caspase-9* (*iCas9*) suicide gene is often used for such purposes as it triggers cell death within minutes following induction. Indeed, *iCas9* triggers apoptosis following activation with a chemical inducer of dimerization (CID) and allows to shut off transgene expression rapidly after CID exposure [59,60].

### 2.3. Conclusions

A major advantage of the regulation of transgene expression by external stimuli is that the induction can be stopped in case of adverse events. However, these regulation systems still suffer from many drawbacks. Indeed, the inducers can generate potential side effects after several exposures. Moreover, for clinical applications, the inducer molecule allowing sensitive promoter activation should be externally provided to the patient and/or by the physician, generating inconvenience. Consequently, there is a need for gene expression switches that are responsive to endogenous markers and that interface directly with e.g., host metabolism in order to trigger adapted responses.

## 3. Internal Induction by Endogenous Stimuli

In their natural ecosystem, cells permanently sense their environment and modulate their behavior in response after integration of inputs signals, such as molecules, temperature, pH, etc. Their sensing capacities can be used and enhanced either by exploiting the natural regulation mechanisms or by creating fully synthetic signaling pathways.

### 3.1. Small Molecule Sensitive Promoters

Endogenous promoters are regulated by transcription factors that can be specifically active or present under specific conditions. For instance, some promoters contain regulatory elements that allow gene expression only in specific cell types, such as neurons, immune cells or hepatocytes [16]. By placing the transgene under the control of these promoters, tissue-specific expression can be achieved albeit this will not be temporally controlled. Interestingly, some tissue-specific promoters have been developed so as to achieve Tet-regulated transgene expression in specific cell types, such as neurons [61,62], adipocytes [63], and endothelial cells [64].

To perform temporal regulation of transgene expression depending on physiological cues, several types of artificial promoters that can respond to defined small soluble factors or to cellular contacts have now been engineered. Inflammation-sensitive promoters derived from endogenous promoters have focused much attention as they can be used to respond to pro-inflammatory molecules, which are commonly associated to many diseases, such as autoimmune diseases. Promoters derived for the *CXCL10* [65,66], *E-selectine (ESELp)* [67], and *IL-1/IL-6* [68] promoters have been designed to induce secretion of effector molecules, mainly anti-inflammatory cytokines, after endogenous signaling in response to inflammatory signals, such as LPS or TNF-α. Similarly, an inducible nuclear factor κB responsive promoter was developed [69]. The CD40L proximal promoter sequence has also been incorporated in LVs to induce transgene expression following T cell activation [70,71]. However, it is clear that with this kind of inducer molecules, the transgenes might also be expressed in response to normal protective inflammation against pathogens. Consequently, promoters that are active specifically in disease conditions should be implemented to control therapeutic transgenes after LV transduction with minimal side effects. For instance, *Saa3* and *Mmp13* promoters are significantly induced in arthritic environment and have been used to regulate the expression of an anti-inflammatory cytokine, e.g., IL-10, showing a benefit on disease outcome [72].

### 3.2. Synthetic Biology Approaches

Synthetic biology is a novel field of research that aims at assembling biological parts into well-organized networks to program novel cellular responses, and hence to widen cellular endogenous therapeutic capabilities. Although promoters derived from endogenous-responsive promoters can be incorporated in LVs to control transgene expression, whole synthetic circuits with promoters that are fully or not orthogonal to cell signaling have been designed to reprogram cellular responses with minimal cross-talks with endogenous signaling. Indeed, the ultimate goal of such approaches is to generate therapeutic responses with an increase efficiency compared to the native responses, in a way which is temporally and spatially restricted by the disease, thus also minimizing side effects.

In order to obtain a functional synthetic circuit, different elements must be integrated (Figure 2). An input or “inducing signal” that is characteristic of the disease, such as a pathological antigen or a dysregulation of the microenvironment, must be recognized by a specific sensor. The inducing signals triggering the synthetic circuit can be of different types. Notably, for endogenous signals that are present in the body, the system can be autonomously detected [73,74]. This approach remains the long-term goal of synthetic immunology. Alternatively, the signal may also be controlled by the user, which offers the advantage of rapid shut down of the circuit if there are complications (see above). Several sensors have been implemented for signal recognition in synthetic networks. First, natural membrane receptors, such as cytokine receptors (TNF for instance [75,76]), or promoters sensitive to pathological signals (NF κB [77]) can be used either directly or after modifications. Furthermore, one of the most promising and widely used strategies to obtain specific sensors is to create chimeric synthetic receptors. The best known and successful synthetic receptor is the chimeric antigen receptor (CAR), which has been primarily used for cancer immunotherapy approaches. Such receptors recognize specific antigens through single chain variable fragments (scFvs) that are derived from specific antibodies. These fragments are fused to the native domains of T cell receptors (TCRs), the ζ signaling domain (signal 1) and the costimulatory receptor (signal 2) that are required for T cell activation [78]. Since the development of CARs, other complex synthetic extracellular sensors have been developed (reviewed here [79]), such as the synthetic notch (SynNotch) and modular extracellular sensor architecture (MESA) or generalized extracellular molecule sensor (GEMS) platforms [80,81,82,83,84]. Signal recognition by such receptors initiates a transducing cascade leading to processing and integration of information (signal type, signal strength, presence of additional signals). This transduction cascade is genetically encoded within complex regulatory circuits. Such transduction circuits can either exploit endogenous signaling cascades (non-orthogonal response) or exploit an exogenous synthetic genetic circuit, which is externally provided in the LV construct for example (orthogonal response). This leads to initiation of cellular effector responses that can be diverse, depending on the targeted pathology or application. For instance, cell death can be triggered by apoptosis or, on the contrary, cell proliferation can be induced. Alternatively, the circuit output can also consist in the secretion of therapeutic molecules, such as antibodies or cytokines.

Several criteria need to be fulfilled to create a reliable synthetic circuit. First, the recognition of input(s) should be specific. To refine such recognition, the Boolean logic of integration of various signals can be exploited (Figure 2, right). Boolean logic gates are designed to integrate multiple inputs and then generate outputs based on logic rules. Synthetic Boolean logic gate circuits can be used to discriminate various parameters such as cell types or disease specificity through the recognition of multiple biomarkers [85]. Several receptors can be multiplexed in order to create more complex circuits. For example, the accuracy of recognition of a pathological condition can be achieved through the combined recognition of two inducing signals that are both necessary for the activation of the effector response: this is an AND loop. For instance, a circuit with such a Boolean logic gate was designed in order to trigger the expression of anti-inflammatory cytokines (IL-4 and IL-10) only in the presence of inflammation associated with psoriasis [75]. Indeed, the levels of two pro-inflammatory cytokines associated with psoriasis (TNF-alpha and IL-22) can be detected using a receptor-based AND logic gate. Likewise, using LVs to engineer immune cells, an AND gate was developed to discriminate tumor cells from healthy cells and hence, to precisely activate CAR T cells against tumor cells [86]. In addition to improving the recognition of a pathological state, these logic gates can also be used to add security checks on the system, via the implementation of feedback loops, which is also an important criterion for synthetic networks [86]. In addition, cells modified with synthetic circuits should also circulate, survive and be functional in pathological conditions, which is not that trivial especially when manipulating immune cells. Nevertheless, the development of synthetic biology approaches represents major advances in novel biotherapies and has a huge potential for personalized medicine therapies, since it can greatly expand the repertoire of detectable soluble molecules and since it is highly programmable in terms of both signal and effector molecules. Of note, the implementation of synthetic biology remains dependent on limitations in the amount of genetic information that can be packaged in LVs.

## 4. Applications of Inducible LVs

### 4.1. Gene Function

Inducible systems for transgene expression following lentiviral transduction have enabled to investigate gene functions both in vitro and in vivo, through approaches allowing to overexpress or conversely to knock-down (KO) the gene of interest, and to monitor the changes that are induced [87,88,89].

While the properties of many genes have been studied through disruption of gene loci in embryos, such approaches are not feasible for genes that are required for embryogenesis or, more generally, that affect cell viability and whose deletion would lead to fetal mortality. However, these limitations can be circumvented by using inducible LVs to trigger complete KO through the expression of single guide RNAs (sgRNAs) after embryonic development [87,90]. Alternatively, for specific applications, the gene of interest should not be permanently shut down but only transiently induced. This can be achieved by using inducible LV systems that allow to finely tune gene silencing by regulation of small interfering or short hairpin RNA (siRNA/shRNA) expression [88,89,91,92]. The main advantage of these systems is that the silencing is fully reversible [91]. Of note, doxycycline-inducible LVs for gene silencing have been extensively used to screen candidate genes potentially implicated in various biological processes, including oncogenesis and immune responses [93,94,95,96,97,98]. On the opposite, inducible LVs have been used to temporary express gene(s) after doxycycline or tetracycline exposure using the tet-ON system in order to study the role of defined proteins [99,100,101].

### 4.2. Cell Reprogramming

Inducible LVs have proven to be useful to reprogram cells either toward a more pluripotent state or, conversely, toward a more differentiated state. Indeed, over-expression of proteins with a well-known effect, such as differentiation factors in cardiac stem cells led to their differentiation in cardiac muscle cells [102]. Complete lineage switches have also been generated, like conversion of fibroblasts to hepatocyte-like cells [103]. Conversely, overexpression of specific factors using inducible LVs can also completely stop differentiation and reprogram cell toward a more pluripotent state [104]. Initially, high expression of multiple transcription factors was performed by combination of several LVs encoding reprogramming factors under constitutive promoters to generate induced pluripotent stem cells (iPSCs). This was resulting in heterologous populations with different potency status, holding back clinical translation. To overcome this hurdle, multicistronic vectors encoding all reprogramming factors with IRES or 2A peptide sequences in between were then designed to trigger iPSCs. With such vectors, the reprogramming remains permanent, due to constitutive transcription factor expression. In the later version of reprogramming LVs, systems for controlling the expression of reprogramming transgene(s) were implemented using either the Tet [105,106,107] or the Cre/LoxP technologies [57,107,108]. The advantage of the Cre/LoxP system for this particular application is that derived iPSCs cells become free of exogenous transgenes. Importantly, while iPSCs generated with inducible LVs retain their pluripotent phenotype when the transgenes are expressed, they are able to differentiate when the ectopic reprogramming genes are switched off [25,109]. Secondary iPSCs could be generated by doxycycline exposure after in vivo differentiation of primary iPSCs [106]. Although some progresses remain to be done before translation to the clinic, induced pluripotent stem cells exhibit a huge potential for regenerative medicine.

### 4.3. Transient Therapeutic Transgene Expression

Lentiviral vectors encoding a regulatable transgene are proving to be useful to help refining disease diagnosis, treatment, and prevention.

One of the most straightforward applications of inducible LVs for therapy is the in vivo-regulation of therapeutic molecule expression. For instance, the systemic administration into SCID mice of LVs encoding a *human clotting Factor IX* (*hFIX*) transgene under a tetracycline-regulated promoter allowed to secrete physiologically-relevant transgenic hFIX doses in the plasma [110]. Doxycycline-regulatable LVs allowing the expression of Fas ligand [111] and GDNF [112] showed interesting beneficial outcome in mouse models of rheumatoid arthritis and Parkinson disease, respectively. Likewise, doxycycline-dependent expression of cytidine deaminase (CDD) could be transiently triggered in HSCs transduced by LVs protecting cell from CDD-induced myelotoxicity and lymphotoxicity [113,114].

Transient transgene activation is also desirable for some therapeutic applications since, after cure, expression of the therapeutic gene may not be needed anymore. Of greater concern, expression of the therapeutic transgene may also become pathological, as it is the case for bone or cartilage repair therapies that often relies on relatively short-term treatment until restoration. To achieve temporally-controlled therapeutic transgene expression for such defects, the Tet system has been implemented in LVs for transduction of mesenchymal stem cells [115] or chondrocytes [116] in order to control amelogenin or Bone Morphogenetic Protein 2, respectively.

Alternatively, other gene regulation approaches have been considered to stop transgene expression once the repair process is achieved. For instance, inducible suicide genes that are incorporated in LV constructs can be induced after repair in order to kill transduced cells and, consequently, to indirectly suppress transgene expression [59,117].

### 4.4. Safety Switches

Inducible suicide genes also present a particular interest for clinical safety purposes. Indeed, the possibility to trigger suicide of LV-transduced cells in case of adverse events is crucial, notably owing to the risk of insertional mutagenesis that is intrinsically linked to this technology. Furthermore, transfer of genetically engineered cells such as CAR T cells can generate complications, like, e.g., cytokine storms or graft-versus-host disease. For all these reasons, it is essential to be able to eliminate dysregulated engineered cells. For instance, the iCas9 suicide gene was inserted in T cells transduced by LVs that also expressed genes encoding a TCR specific for leukemia-associated minor H antigen (HA-1) and a CD8 coreceptor [118]. After in vitro exposure to the iCas9 CID, cell death can be rapidly observed (in few minutes) upon iCas9 dimerization. Moving a step beyond, T cells were transduced with a cassette encoding iC9 before transfer into patients that had previously undergone a stem cell transplantation [60,119]. This clinical trial showed tremendous results, as CID infusion led to eradication of 90% of iC9-expressing cells within 2 h and allowed to control GVHD [119]. Such suicide switches will surely become more and more used for cell therapies, notably those involving T cells. Alternatively, CAR genes can be inserted under the control of doxycycline-sensitive promoters, hence creating another possibility to stop the induction in case of adverse events [120,121].

## 5. Conclusions and Perspectives

Several systems have been implemented in LVs for achieving inducible transgene expression. Although externally-provided cues can be used to control transgene expression through specific promoters, the development of completely autonomous inducible systems is the focus of intense research efforts currently. In this direction, it should be noted that CAR T cells efficacy can be potentiated by combining the direct attack of the CAR T cells themselves along with the secretion of ectopic cytokines in order to boost the immune response against target components (CAR T fourth generation, TRUCK CAR) [122,123]. This is achieved by inserting cytokine transgenes under the control of a Nuclear Factor of Activated T-cells (NFAT)-inducible promoter, since CAR stimulation induces, phosphorylation of NFAT, which subsequently migrates to the nucleus and activates sensitive promoters inserted in the LV construct and hence, gene expression.

Synthetic immunology is an extremely promising research area in order to induce self-regulation of effectors upon in vivo sensing of defined biomarkers. In addition, since the inputs and the outputs of synthetic circuits can easily be programmed, the effector molecule(s) expression might be adapted for each patient, leading to much-wanted personalized medicine approaches. Such approaches would be particularly useful for the treatment of chronic diseases, as upon disease flares, the expression of therapeutic molecule would be immediately re-induced, ideally before the appearance of symptoms. Alternatively, for the treatment of chronic diseases, clock-regulated systems dependent on the circadian rhythm might also become a scientific reality of the next decades for achieving a rhythmic expression of desired therapeutic transgenes [124]. Going further, upon thorough safety assessments, such synthetic circuits could be used for prevention in people at risk to develop certain diseases, such as cancers or autoimmune diseases. However, many progresses remain to be achieved especially in terms of safety and orthogonality, as cross-talks with endogenous signaling cascade might occur.

This review focused on transcriptional transgene regulation, for which promoters responsive to defined cues are controlled by transcription factors; However, gene expression can also be regulated at several levels. Indeed, by incorporating specific regulatory sequences, a controlled alternative splicing may result in the expression of different protein forms. For instance, a LV allowing the expression of a membrane-anchored vs. secreted immunoglobulin depending on B cell maturation status has been designed from the same construct [125]. At the translational level, ribozymes or riboswitches can be engineered so as to respond to specific triggers and to control protein translation [126].

Alternatively, LVs might be used to insert transgenes at defined genomic locations in order to benefit from the endogenous regulation of these gene loci. For instance, two LVs that express the Cre recombinase and a transgene adjacent to LoxP site, were developed and led to transgene insertion in LoxP or pseudo-LoxP sites present in the genome [127]. Nanoblades, a combo system engineered from retrovirus particles, was developed to facilitate infusion into cells of sgRNA and Cas9 proteins, and has proven to be a versatile device for genome editing [128]. This system can introduce double stranded-breaks at defined locations and, if a DNA repair matrix is provided, the transgene would be inserted inside the break and thus would be regulated by surrounding sequences.

Finally, it is clear that the on-going development of many new technologies will widen the approaches aimed at achieving inducible transgene expression. For instance, using optogenetics with light-controlled promoters, transgene expression from implanted cells can be regulated by brain signals [129]. Alternatively, smartphone-switched devices have been developed to control in vivo the secretion of glucagon or insulin from implanted cells through wireless communication [130]. Such smartphone-controlled transgene approaches might become a reality in the coming years in a more and more connected society.

## Figures and Tables

**Figure 1 viruses-12-01427-f001:**
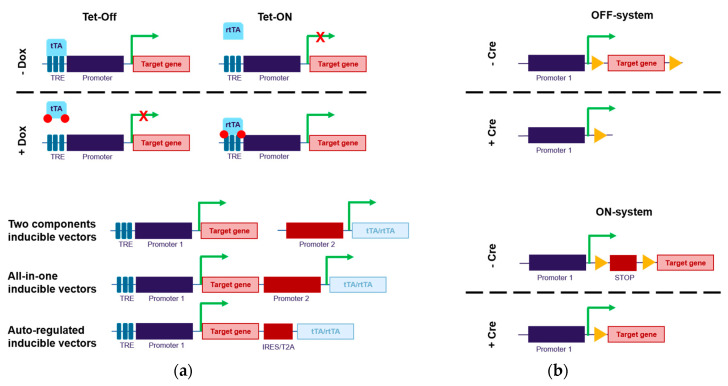
Externally inducible promoter principle. (**a**) The TET-system. For the tet-ON version, upon addition of doxycycline, the tetracycline controlled trans-activator (tTA) can no longer bind Tet response element (TRE), hence leading to interruption of gene expression, whereas for the tet-OFF version, reverse tetracycline controlled trans-activator (rtTA) is not bound to TRE in absence of doxycycline, which does not allow gene transcription until addition of doxycycline. Several strategies have been tested for implantation of the Tet-system in lentiviral vectors (LVs). Initially, two vectors, one encoding the target gene under the control of a TRE and another encoding the trans-activator or the trans-repressor were used. These two parts were then combined into a single vector. Moving to a step further to reduce background level, the trans-activator/trans-repressor were also placed under the regulation of TRE. (**b**) The LoxP/Cre system. For this system, two versions have also been tested in LVs. The gene of interest can be placed between two LoxP sites. Consequently, upon Cre exposure, it will be excised and hence stop transgene expression. Alternatively, a gene silencing sequence (stop codon, poly(A) signal) is inserted with LoxP sites around, just before the target gene. Consequently, the addition of Cre recombinase will lead to the removal of the stop codon and lead to transgene expression.

**Figure 2 viruses-12-01427-f002:**
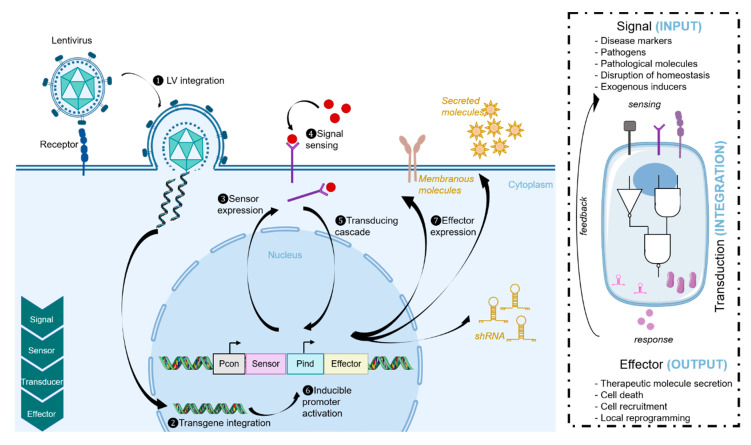
Synthetic biology approaches. Lentiviral vectors are used to transduce cells. These vectors encode the main parts of the synthetic network: receptors, transducers, and effectors. Inputs signals that are markers of a pathological condition will be sensed by specific membranous or intracellular receptors, inducing a transducing cascade. This signal integration will lead to the expression of effector molecules such as shRNA, cytokines, or pro-apoptotic cues.

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
