# Peer review of "Toward Tightly Tuned Gene Expression Following Lentiviral Vector Transduction"

_viruses, 2020, doi:10.3390/v12121427_

Round 1
Reviewer 1 Report
The authors review the literature looking specifically at promoters used in lentiviral vectors that can be tightly controlled by endogenous or exogenous stimuli and conclude that expression control is important for gene therapy.
This reviewer only has minor points:
- Please clarify sentence in line 38
- Pseudotyping choice to restrict expression to certain cell types could be strengthen with examples (line 45)
- Please consider rephrasing sentence in line 57
- rtTA should be tTA (Line 134)
- It would be important to expand on the variants of tet-responsive promoters (lines 139-141)
- Please state the downside(s) of using common ingestible compounds to control promoters (lines 183-186)
- The authors could expand on the details of the promoters derived from endogenous promoters to explain to the readers their molecular mechanistic control (lines 264-266)
- Please provide references for your statement regarding the ease of gene function study by inducible systems (lines 344-347)
- The authors should include other systems to control promoters, such as ionizing radiation that can control the EGR1 promoter.
Author Response
Replies to Reviewer 1
The authors review the literature looking specifically at promoters used in lentiviral vectors that can be tightly controlled by endogenous or exogenous stimuli and conclude that expression control is important for gene therapy.
This reviewer only has minor points:
- Please clarify sentence in line 38
(Reply) As suggested by this Reviewer, a term has been removed line 38 “(and quiescent cells)”. Indeed, to clarify the distinction between quiescent and non-dividing cells, we consider that non-dividing cells might be quiescent, meaning they are able to divide again, while there are other non-dividing cells such as apopotic or necrotic cells.
- Pseudotyping choice to restrict expression to certain cell types could be strengthen with examples (line 45)
(Reply) We thank this Reviewer for his/her suggestion and we have introduced examples and references in lines 45-47.
- Please consider rephrasing sentence in line 57
(Reply) The sentence has been reformulated as follow: “This deletion minimizes activation of genes located at close proximity of the vector integration site in the genome.” (line 59).
- rtTA should be tTA (Line 134)
(Reply) We apologize for this mistake, which has been corrected line 137.
- It would be important to expand on the variants of tet-responsive promoters (lines 139-141)
(Reply) We have taken this remark into account. Some precisions regarding Ptet variants have been added in lines 144-146: “expression either by truncating the CMV minimal promoter in Ptet, by using the HIV-1 long terminal repeat promoter instead of CMV in Ptet, or by random mutations in Ptet”
- Please state the downside(s) of using common ingestible compounds to control promoters (lines 183-186)
(Reply) Drawbacks of common ingestible compounds as inducer of regulated promoters are now listed in lines 189-191.
- The authors could expand on the details of the promoters derived from endogenous promoters to explain to the readers their molecular mechanistic control (lines 264-266)
(Reply) We agree that the molecular mechanisms are not described, however often these mechanisms are not known but globally there are expected to be the same as the ones that are regulating the endogenous promoters because the same transcription binding domains are present on the ectopic construct. Thus we added “after endogenous signaling” in lines 278-279.
- Please provide references for your statement regarding the ease of gene function study by inducible systems (lines 344-347)
(Reply) We apologize for these misleading terms. It is not a matter of ease in terms of technical feasibility but more in terms of possible modifications (such as KO that would be deleterious if done before development). Consequently, we removed the terms “easily” to avoid confusion.
- The authors should include other systems to control promoters, such as ionizing radiation that can control the EGR1 promoter.
(Reply) We thank this Reviewer for this excellent suggestion. Such regulated systems are now mentioned lines 204-207: “Alternatively, ionizing radiations have been shown to activate of the early growth response gene (EGR1) promoter via a consensus sequence. This opens the path for transgene activation at the radiation field to enhance treatment efficacy by local co-delivery of therapeutic molecules”.
Reviewer 2 Report
This Review by Page and co-workers is focused on an interesting aspect of the setting up of lentiviral vectors, i.e. the possibility to modulate transgene expression by adopting different promoters. The review is well written and organized and the Figures are sufficient in numebr and explicative. The only minor point I want to rise is the lack of references supporting the concepts described in the text, especially in the initial paragraphs. As an expample:
Page 2, lines 44-45: "Conversely, by carefully choosing the pseudotyping virion surface glycoproteins, vector tropism may also be restricted to certain cell types" it would be important to know whether this has been demonstrated (and thus a reference should be introduced) or whether this is a comment of the Authors based on their personal experience
Page 2, lines 59-61: "Several promoters can be combined to express multiple transgenes from a single vector although this is limited by the 60 packaging capacity of LVs (around 10 kb) and although this can lead to promoter interferences" same as above
Other than that, the review is absolutely worth to be published in Viruses.
Author Response
Replies to Reviewer 2
This Review by Page and co-workers is focused on an interesting aspect of the setting up of lentiviral vectors, i.e. the possibility to modulate transgene expression by adopting different promoters. The review is well written and organized and the Figures are sufficient in number and explicative. The only minor point I want to rise is the lack of references supporting the concepts described in the text, especially in the initial paragraphs. As an example:
Page 2, lines 44-45: "Conversely, by carefully choosing the pseudotyping virion surface glycoproteins, vector tropism may also be restricted to certain cell types" it would be important to know whether this has been demonstrated (and thus a reference should be introduced) or whether this is a comment of the Authors based on their personal experience
(Reply) As requested by this Reviewer, some precisions and references have been added regarding tropism restriction by pseudotyping, especially with targeting antibodies, which has already been published (lines 45-47).
Page 2, lines 59-61: "Several promoters can be combined to express multiple transgenes from a single vector although this is limited by the packaging capacity of LVs (around 10 kb) and although this can lead to promoter interferences" same as above
(Reply) We thank this Reviewer for helping us to clarify these points. References dealing with promoter cross-talks have been added, especially the paper of Park et al 2019 (Ref 15), which compared several organization of promoters in order to reduce interferences.